# Search and Identification of Amyloid Proteins

**DOI:** 10.3390/mps6010016

**Published:** 2023-02-04

**Authors:** Tatyana A. Belashova, Anna A. Valina, Evgeniy I. Sysoev, Maria E. Velizhanina, Andrew A. Zelinsky, Alexey P. Galkin

**Affiliations:** 1St. Petersburg Branch, Vavilov Institute of General Genetics, Universitetskaya Emb. 7/9, 199034 St. Petersburg, Russia; 2Laboratory of Amyloid Biology, St. Petersburg State University, Universitetskaya Emb. 7/9, 199034 St. Petersburg, Russia; 3Department of Genetics and Biotechnology, St. Petersburg State University, Universitetskaya Emb. 7/9, 199034 St. Petersburg, Russia; 4Laboratory of Signal Regulation, All-Russia Research Institute for Agricultural Microbiology, Podbelskogo 3, 196608 St. Petersburg, Russia

**Keywords:** amyloids, prions, proteomic screening, fibril immunoprecipitation, detergent resistance, mass spectrometry, electron microscopy, Congo red

## Abstract

Amyloids are fibrillar proteins with a cross-β structure. Pathological amyloids are associated with the development of a number of incurable diseases, while functional amyloids regulate vital processes. The detection of unknown amyloids in living objects is a difficult task, and therefore the question of the prevalence and biological significance of amyloids remains open. We present a description of two methods, the combination of which makes it possible to find and identify amyloid proteins in the proteome of various organisms. The method of proteomic screening for amyloids allows the detection of the proteins that form SDS-resistant aggregates. SDS resistance is a general feature of amyloid fibrils. Protein aggregates resistant to SDS treatment can be collected by ultracentrifugation and further identified by mass spectrometry. However, in addition to amyloids, SDS-resistant aggregates contain some non-amyloid proteins. To test the amyloid properties of proteins identified by proteomic screening, we developed the method of fibril immunoprecipitation followed by Congo red staining and birefringence analysis. The methods of proteomic screening and immunoprecipitation of fibrillar proteins have been successfully tested and applied for the identification of amyloid proteins in yeast and vertebrates.

## 1. Introduction

Amyloids are fibrillar proteins with a cross-beta structure. The formation of cytotoxic amyloid fibrils is often associated with the development of incurable systemic and localized amyloidosis [1]. At the same time, the so-called functional amyloids are characterized. This group includes proteins that are normally stored or function in the form of amyloid fibrils [2,3]. The search for and identification of proteins that form amyloid fibrils in various living objects is a rather difficult task. Most of the pathological and functional amyloids that have been characterized to date have been identified in the studies of a particular protein or a specific pathology. This approach does not allow assessment of the prevalence and biological significance of the pathological and functional amyloids in nature.

We have developed and successfully tested a PSIA-LC-MALDI method that detects proteins with amyloid properties in the proteomes of various organisms. The proposed method is based on the general feature of all known amyloid fibrils—their resistance to treatment with detergents such as SDS [4,5]. In contrast to amyloids, most non-amyloid protein aggregates and complexes are disassembled to monomers upon treatment with 1% SDS at room temperature. High molecular weight amyloid fibrils after SDS treatment can be separated from other proteins by ultracentrifugation. The proteins contained in this fraction are identified by mass spectrometry. The PSIA-LC-MALDI has been successfully used to identify proteins that form amyloid-like aggregates in yeast, bacteria, plants and vertebrate brain neurons [5,6,7,8,9,10]. It should be noted that some non-amyloid protein aggregates and complexes are also resistant to SDS treatment. Considering this, it is necessary to check whether the proteins identified in proteomic screening really form amyloid fibrils in vivo. Various techniques are used to test the amyloid properties of fibrillate proteins. In particular, the presence of a cross-beta structure, which is characteristic of amyloids, can be assessed using solid-state NMR [11], X-ray diffraction [12] or cryo-electron microscopy [13]. Unfortunately, in most cases, these methods apply only to the analysis of the purified protein in vitro. The customary way of evaluating protein amyloid properties in living organisms is to stain cytological preparations with amyloid-specific dyes, such as thioflavin T, thioflavin S or Congo red, and search for the colocalization of these dyes with the protein-specific antibodies [8,14,15]. In rare cases, these dyes can bind not just amyloid proteins [16,17,18]. Nowadays, staining of cytological preparations with Congo red, followed by the analysis of birefringence during polarization, is considered the most convincing proof of the amyloid nature of a protein [15]. Amyloids stained with Congo red exhibit a yellow-green birefringence in polarized light. At the same time, colocalization of the studied protein with an amyloid-specific dye cannot be final proof of its amyloid nature. Many proteins can be localized in the same area of a cytological preparation, and at such a level of resolution, it is impossible to conclude which of them binds Congo red. We have developed a method for the isolation of native amyloid fibrils from living organisms using immunoprecipitation for further evaluation of their amyloid properties ex vivo [8,19]. This method includes immunoprecipitation of the fibrillar protein detected in proteomic screening followed by the release of the protein from magnetic beads without destroying the fibrils. Fibrils are detected using electron microscopy and also stained with Congo red to assess birefringence.

The combination of methods of proteomic screening and analysis of amyloid properties of fibrillar proteins ex vivo enables the discovery of infectious amyloids (prions) and non-infectious amyloids in any organism. Using these approaches we identified functional amyloid FXR1 in mammalian brain neurons [5] and obtained direct evidence of the amyloid properties of yeast prions [19]. The development of these methods opens prospects for a systematic search for uncharacterized functional and pathological amyloids in various organisms.

## 2. Experimental Design

### 2.1. Proteomic Screening for Amyloid Proteins

The method of proteomic screening of proteins that form amyloid-like aggregates (PSIA-LC-MALDI) includes several steps (Figure 1). First, the protein lysate isolated from the tissues or unicellular organism is treated with 1% SDS at room temperature. This treatment facilitates the disassembly of most aggregates and complexes, while amyloid protofibrils remain intact and are not disassembled into monomers. At the next step, high molecular weight protein aggregates resistant to SDS treatment are separated from other proteins by ultracentrifugation. After washing and repeated ultracentrifugation, SDS is removed, and the samples are treated with trypsin. Peptides formed as a result of trypsinolysis are separated by HPLC and then identified by time-of-flight mass spectrometry. Based on these results, the mass spectrometer compiles a list of proteins present in the studied fraction.

The disadvantage of this method is that, in addition to amyloid proteins, other proteins that form SDS-resistant aggregates are also identified [6]. This disadvantage can be overcome if the analysis is carried out parallel to “a negative control”. For example, a comparative study of proteins detected in proteomic screening in patients with suspected amyloidosis and patients of the control group can reveal proteins that aggregate only in pathology. For a search of functional amyloids in different organisms, the concentration of the detergent can be modified. For example, 2% SDS can be used instead of 1% for a more stringent selection of candidate proteins.

### 2.2. Immunoprecipitation of Amyloids and Ex Vivo Fibril Staining

To analyze the amyloid properties of proteins identified by proteomic screening, we developed an approach for isolating native fibrils from living organisms. At the first step, primary antibodies specific to the protein of interest are immobilized onto magnetic beads covered with protein A or G (Figure 2). The binding specificity of A or G proteins to various primary antibodies has to be determined according to the manufacturer’s protocol. At the next step, the protein lysate is incubated with protein A- or G-coupled magnetic beads. The magnetic particles are washed to get rid of unbound antibody material. In order to analyze the amyloid properties of the protein of interest, it is necessary to gently separate it from antibodies and magnetic particles, avoiding boiling. The resulting proteins are precipitated by ultracentrifugation and resuspended in water. Then, the fibrillar structure of the studied proteins can be analyzed using electron microscopy. Additionally, protein samples isolated by immunoprecipitation are stained with Congo red to analyze birefringence. This approach allows identification of both functional and pathological amyloids.

The disadvantage of this method is that in addition to the target protein, other proteins bound to amyloid fibrils are precipitated. That circumstance and the low amount of the precipitated protein complicate the further analysis of fibrils by physical methods that are used in the study of amyloids in vitro. Theoretically, after immunoprecipitation, the fibrils can be incubated with 1% SDS and precipitated by ultracentrifugation to eliminate other protein contaminants. In this case, it is necessary to isolate a large amount of the target protein so as not to lose it during additional treatments.

### 2.3. Materials

Deionized water.Distilled water.TBS (see Section 5).PBS (see Section 5).Lysis buffer 1 (see Section 5).Lysis buffer 2 (see Section 5).PMSF (Sigma-Aldrich, St. Louis, MO, USA, cat. no. P7626).DTT (Thermo Fisher Scientific, Waltham, MA, USA, cat. no. R0861).EDTA (Sigma-Aldrich, St. Louis, MO, USA, cat. no. ED2SS).Tris base (Sigma-Aldrich, St. Louis, MO, USA, cat. no. T1503).NaCl (Sigma-Aldrich, St. Louis, MO, USA, cat. no. S9888).KCl (Sigma-Aldrich, St. Louis, MO, USA, cat. no. P3911).KH_2_PO_4_ (Sigma-Aldrich, St. Louis, MO, USA, cat. no. P5379).Na_2_HPO_4_ (Sigma-Aldrich, St. Louis, MO, USA, cat. no. S9763).Complete Protease Inhibitor Cocktail ×100 (Roche, Basel, Switzerland, cat. no. 78429, store at –20 °C).Glass beads (Cole-Parmer Instrument Company, Vernon Hills, IL, USA, cat. no. 11079105).Liquid nitrogen.Ice.RNAse A (Thermo Fisher Scientific, Waltham, MA, USA, cat. no. EN0531, store at −20 °C).SDS (VWR International, Radnor, PA, USA, cat. no. 0227).Sucrose (Sigma-Aldrich, St. Louis, MO, USA, cat. no. S0389).25% sucrose-TBS cushion (see Section 5).Tween-20 (Sigma-Aldrich, St. Louis, MO, USA, cat. no. P1379).Formic acid (Sigma-Aldrich, St. Louis, MO, USA, cat. no. 33015).2-Mercaptoethanol (Sigma-Aldrich, St. Louis, MO, USA, cat. no. 63690).Bromophenol Blue (Sigma-Aldrich, St. Louis, MO, USA, cat. no B0126).SDS-PAGE loading buffer (see Section 5).Ammonium bicarbonate (Sigma-Aldrich, St. Louis, MO, USA, cat. no. 09830).Iodoacetamide (Sigma-Aldrich, St. Louis, MO, USA, cat. no. I1149).Trypsin (Sigma-Aldrich, St. Louis, MO, USA, cat. no. T4174, store frozen between −10 and −40 °C).Trifluoroacetic acid (TFA) (Sigma-Aldrich, St. Louis, MO, USA, cat. no. 302031).Acetonitrile (Sigma-Aldrich, St. Louis, MO, USA, cat. no. 34851).An amount of 37% HCl (Sigma-Aldrich, St. Louis, MO, USA, cat. no. 320331). CAUTION: Toxic when inhaled, causes irritation to the respiratory tract and causes skin burn.Urea (Sigma-Aldrich, St. Louis, MO, USA, cat. no. U5128).Thiourea (Sigma-Aldrich, St. Louis, MO, USA, cat. no. T7875).Calibration Standard for mass spectrometer (Bruker Daltonics, Billerica, MA, USA, cat. no. 8222570, model Peptide Calibration Standard II, store at less than 0 °C).α-cyano-4-hydroxycinnamic acid (Bruker Daltonics, Billerica, MA, USA, cat. no. 8201344, store at 2–8 °C).Magnetic beads covered with protein A or G (Sileks, Moscow, Russia, cat. no. K0181 or K0182, model SileksMag-Protein A or SileksMag-Protein G, respectively).Anti-target serum or antibodies (able to recognize the native protein and suitable for immunoprecipitation).Binding buffer (see Section 5).Elution buffer (see Section 5).Glycine (ICN Biomedicals, Costa Mesa, CA, USA, cat. no. 194681).Neutralizing buffer (see Section 5).Congo Red dye (Thermo Fisher Scientific, Waltham, MA, USA, cat. no. AAB2431014).Uranyl acetate (Electron Microscopy Sciences, Hatfield, PA, USA, cat. no. 22400).

### 2.4. Equipment

Pipettes (HTL lab solutions, Warsaw, Poland, cat. no. 7901, model Discovery Comfort).Microcentrifuge Tubes, 0.2 mL (Axygen, Corning, NY, USA, cat. no. PCR-02-C, model Thin-Wall PCR Tubes with Flat Caps, Clear).Microcentrifuge Tubes, 0.5 mL (Axygen, Corning, NY, USA, cat. no. PCR-05-C, model Thin-Wall PCR Tubes with Flat Caps, Clear).Microcentrifuge Tubes, 1.5 mL (SSIbio, Lodi, CA, USA, cat. no. 1210-00).Microcentrifuge Tubes, 2.0 mL (Axygen, Corning, NY, USA, cat. no. MCT-200-C, model MaxyClear Snaplock Microcentrifuge Tube, Polypropylene, Clear).Ultracentrifuge Tubes, 230 µL (Beckman Coulter Life Sciences, Brea, CA, USA, cat. no. 343621, model Open-Top Thickwall Polypropylene Tube).Bulk Pipette Tips, 200 μL (SSIbio, Lodi, CA, USA, cat. no. 4230N00).Bulk Value Pipette Tips, 1000 µL (SSIbio, Lodi, CA, USA, cat. no. 4330-01).Syringe Filter, 0.22 µm (Millipore Sigma, Burlington, MA, USA, cat. no. SLLG033, model Millex Syringe Filter).Conical centrifugal tube, 15 mL (Axygen, Corning, NY, USA, cat. no. SCT-15-R-S).Cryogenic laboratory mill (SPEX SamplePrep, Metuchen, NJ, USA, www.spexsampleprep.com (accessed on 29 October 2022), model 6870 Large Freezer/Mill).Benchtop homogenizer (MP Biomedicals, Santa Ana, CA, USA, cat. no. 116004500, model FastPrep-24).Vortex mixer (Biosan, Riga, Latvia, cat. no. BS-010201-AAA, model FV-2400 Micro-Spin).Refrigerated centrifuge (Thermo Fisher Scientific, Waltham, MA, USA, cat. no. 11175774, model Jouan CR3i multifunction).Sealed Angle Rotor (Thermo Fisher Scientific, Waltham, MA, USA, cat. no. 11840562, model AC 2.14).Polycarbonate tubes, 16 × 76 mm, thick-walled (Beckman Coulter Life Sciences, Brea, CA, USA, cat. no. 355630).Refrigerated ultracentrifuge (Beckman Coulter Life Sciences, Brea, CA, USA, cat. no. 392050, model Optima L-100 XP).Fixed-Angle Rotor (Beckman Coulter Life Sciences, Brea, CA, USA, cat. no. 31066, model Type 75 Ti).ThermoMixer comfort (Eppendorf, Hamburg, Germany, cat. no. 5355000011).Vacuum concentrator (Labconco, Kansas City, MO, USA, cat. no. 7810030, model CentriVap Benchtop Centrifugal Vacuum Concentrator with acrylic lid).Detergent removal columns (Thermo Scientific, Waltham, MA, USA, cat. no. 88305, model HiPPR™ Detergent Removal Spin Columns).Desalting columns (Thermo Scientific, Waltham, MA, USA, cat. no. 45-001-527 Cytiva PD SpinTrapTM G-25 Desalting Columns).HPLC reverse-phase column 150 mm × 75 μm, particle size 5 μm (Thermo Scientific, Waltham, MA, USA, https://www.fishersci.se/shop/products/acclaim-pepmap-c18-300-hplc-column-5-m-particle-size/p-4523544 (accessed on 30 October 2022), cat. no. 163574, model PepMap™ 300 C18 HPLC Column).Nano high-performance nanoflow liquid chromatograph (Thermo Scientific, Waltham, MA, USA, cat. no. ULTIM3000RSLCNANO, model UltiMate™ 3000 RSLCnano System).microtiter plate, 384-sample, with spot diameter 800 µm (Bruker Daltonics, Billerica, MA, USA, https://bruker-labscape.store/collections/maldi/products/maldi-anchorchip-targets?variant=36420251320478 (Accessed on 15 November 2022), cat. no. 8280790, model MTP AnchorChip 384 BC).LC-MALDI Fraction Collector (Bruker Daltonics, Billerica, MA, USA, model Proteineer fc II).Mass spectrometer (Bruker Daltonics, Billerica, MA, USA, model Ultraflextreme MALDI-TOF/TOF).WARP-LC software, version 1.2 (Bruker Daltonics, https://bruker-daltonics-warp-lc.software.informer.com/1.2 (accessed on 1 November 2022)).Mascot version 2.4.2 software (Matrix Science; http://www.matrixscience.com/mascot_support_v2_4.html (accessed on 30 October 2022)).BioTools software, version 3.2 (Bruker Daltonics, https://bruker-daltonics-biotools.software.informer.com/3.2 (accessed on 30 October 2022)).Magnetic rack (Sileks, Moscow, Russia, cat. no. EQRM06, model MagRack 6).Programmable rotator (Biosan, Riga, Latvia, cat. no. BS-010117-AAG, model Multi Bio RS-24).Refrigerated ultracentrifuge (Beckman Coulter Life Sciences, Brea, CA, USA, cat. no. 393315, model Optima MAX-XP).Fixed-Angle Rotor (Beckman Coulter Life Sciences, Brea, CA, USA, cat. no. 343840, model TLA-100).Fine tweezer (Prokit’s Industries, HsinTienCity, New Taipei City, Taiwan, cat. no. 1PK-102T, model Super Fine Tip Straight Tweezer—120 mm).Formvar coated copper grids (Electron Microscopy Sciences, Hatfield, PA, USA, cat. no. 50-260-36 300, model Formvar/Carbon Film 10 nm/1 nm thick on Square 300 mesh Copper Grid).Transmission Electron Microscope (Jeol, Peabody, MA, USA, www.jeol.com (accessed on 29 October 2022) model JEM-2100).Microscope Slides 26 × 76 mm (Deltalab, Barcelona, Spain, cat. no. D100001, D100003, model EUROTUBO^®^ slides).Cover Slides 24 × 24 mm (Menzel Gläser, Braunschweig, Germany, cat. no. MENZBB024024A123).Filter paper (Bio-Rad Laboratories, Hercules, CA, USA, cat. no. 1620118).Polarized light microscope (PZO Microscopy, Warsaw, Poland, pzo.waw.pl (accessed on 29 October 2022), model Biolar PI-PZO).ToupCam camera (ToupTek Photonics, Hangzhou, China, www.touptek.com (accessed on 29 October 2022), model UCMOS10000KPA).ToupView(x86) software, version 3.7 (ToupTek Photonics, http://www.touptek.com/download/showdownload.php?lang=en&id=33 (accessed on 29 October 2022)).

## 3. Procedure

### 3.1. PSIA-LC-MALDI

#### 3.1.1. Preparation of Cell/Tissue Lysates

##### Lysates from Yeast Cells

Suspend the cell mass (grown in 300 mL of suitable medium) on ice in 5 mL of cold Lysis buffer 1 in 15 mL centrifugal tubes.Add an equal volume of glass beads for disruption.Run 10 cycles of destruction of 20 s using benchtop homogenizer. In between cycles of destruction, incubate the samples for 1 min on ice.Transfer the cell lysate into new tubes and centrifuge at 805× *g*, 4 °C for 5 min.Transfer the supernatant (clarified cell lysate) into new tubes.



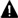

**CRITICAL STEP** Do not freeze lysates! Always work only with freshly prepared lysates.

##### Lysates from Animal Tissue

Homogenize tissue samples using a Cryogenic laboratory mill at −196 °C in liquid nitrogen according to Table 1.

2.Store the homogenized tissue at −70 °C.



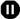

**PAUSE STEP** The homogenized samples can be stored at −70 °C for about 3 years.

3.Suspend 1–1.5 g of homogenized tissue on ice in 6 mL of cold Lysis buffer 1 in 15 mL tubes.4.Centrifuge at 1500× *g*, 4 °C for 10 min.5.Transfer the supernatant into new tubes, add 5 mg of RNase A (75–150 units/mg) and incubate at 30 °C for 15 min.

#### 3.1.2. Isolation of Detergent-Resistant Protein Fractions

6.Gently layer the protein lysate in centrifuge tubes onto 1 mL of 25% sucrose-TBS cushion so that the phases do not mix.7.Ultracentrifuge the applied lysates at 151,000× *g*, 8 °C for 2 h.8.Remove the supernatant, suspend thoroughly the pellet in 1 mL of Lysis buffer 1 and transfer suspension into new tubes. Add 6.2 mL of Lysis buffer 1 and 0.8 mL of 10% SDS solution (final concentration 1% SDS) and mix.



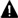

**CRITICAL STEP** Thoroughly suspend the precipitate until homogeneous, free of sediment particles.

9.Gently layer the protein samples in centrifuge tubes onto 1 mL of 25% sucrose-TBS cushion containing 1% SDS. Ultracentrifuge test tubes at 151,000× *g*, 18 °C, for 8 h, with the Delayed Start Program for 8 h at 18 °C.



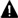

** CRITICAL STEP** Incubation of the lysate with SDS prior to centrifugation is a critical step as it causes disassembly of non-amyloid protein aggregates.

10.Suspend the resulting pellet of detergent-resistant protein aggregates in 1 mL of deionized water, transfer in new centrifuge tube, add water up 8 mL, mix and centrifuge at 151,000× *g*, 8 °C for 2 h.11.Remove supernatant and suspend the pellet in 90 µL of deionized water.



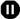

**PAUSE STEP** The resulting solution can be stored at −20 °C for a week.

#### 3.1.3. Proteins Separation by HPLC



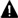

**CRITICAL STEP** Be careful, steps 12–15 are applicable only for yeast cells. For animal tissues, go straight to point 16. Steps 12–15 are not applicable to mammalian tissues because after lyophilization and dissolution in formic acid, an insoluble gel-like precipitate is formed.

12.Lyophilize isolated detergent-resistant protein fractions in the pellet using the vacuum concentrator.13.Dissolve the lyophilized samples in ~120–200 µL of 98% formic acid and treat within 10 min at room temperature.14.Dry the samples until formic acid is completely removed in the vacuum concentrator.15.Solubilize the samples in 90 µL of TBS.16.Add 30 µL of 4× SDS-PAGE loading buffer to the samples and boil at 95 °C for 10 min.



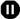

**PAUSE STEP** Samples with denatured proteins can be stored at −20 °C for about month.

17.Remove detergents from the samples using HiPPR Detergent Removal columns, according to the manufacturer’s protocol.18.Remove salts from the samples using Cytiva PD SpinTrapTM G-25 Desalting Columns, according to the manufacturer’s protocol.19.Supplement the final samples (volume 50 μL, total protein concentration 0.2–0.4 mg/mL) with 1 μL of freshly prepared 50 mM DTT in 50 mM ammonium bicarbonate and incubate at 50 °C for 15 min.20.Supplement the samples with 1 μL 100 mM iodoacetamide in 50 mM ammonium bicarbonate and incubate at 20 °C for 15 min in the dark.21.Supplement the samples with 1 μL DTT to inactivate iodoacetamide and 5 μL trypsin (10 ng/μL) and incubate overnight at 37 °C.22.Inactivate the trypsin by adding 0.5 μL 10% TFA followed by centrifuging at 20,000× *g*, 4 °C for 30 min.23.Load the final peptide mixtures (1 μL) onto an Acclaim PepMap 300 HPLC reverse-phase column (150 mm, 75 μm, particle size 5 μm) and separate in an acetonitrile gradient (2–90%) during 45 min using an UltiMate 3000 UHPLC RSLCnano high-performance nanoflow liquid chromatograph.24.Collect peptide fractions every 10 s and load onto a 384-sample MTP AnchorChip 800/384 microtiter plate using spotter Proteineer fc II.

#### 3.1.4. Identification of Proteins

25.Identify peptides using the mass spectrometer. MS spectra for each peptide fraction are determined using the WARP-LC software. The program determines a set of unique peptides characterized by a certain retention time, charge and molecular weight, and also performs MS/MS analysis for these peptides in fractions (spots) with the maximum concentration (peak intensity) of these peptides.26.Correspondence analysis between experimental spectra and corresponding proteins is performed automatically using Mascot software version 2.4.2 in the UniProt database (http://www.uniprot.org (accessed on 15 November 2022)) limited to the organism of interest. α-cyano-4-hydroxycinnamic acid is used as a matrix. When analyzing, use the following “Mass tolerance” parameters: precursor mass tolerance 100 ppm, fragment mass tolerance 0.9 Da. Use Peptide Calibration Standard II 8,222,570 as standard. Carboxymethylation of cysteine, partial oxidation of methionine and one omitted trypsinolysis site should be considered as valid modifications.27.Match the obtained mass spectra to the corresponding proteins using NCBI database.28.Use the BioTools software for manual validation of protein identification.

### 3.2. Immunoprecipitation of Amyloids and Ex Vivo Fibril Staining

#### 3.2.1. Preparation of Cell/Tissue Lysates

##### Lysates from Yeast Cells

Suspend the cell mass (grown in 50 mL of suitable medium) on ice in 400 µL of cold Lysis buffer 2 in 1.5 mL microcentrifuge tube.Add an equal volume of glass beads for disruption.Run 10 cycles of destruction of 20 s using benchtop homogenizer. In between cycles of destruction, incubate the samples for 1 min on ice.Transfer the cell lysate into new tubes and centrifuge at 805× *g*, 4 °C for 5 min.Transfer the supernatant (clarified cell lysate) into new tubes, store at 4 °C.



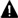

**CRITICAL STEP** Do not freeze lysates! Always work only with freshly prepared lysate.

##### Lysates from Animal Tissue

Homogenize tissue samples using a Cryogenic laboratory mill at −196 °C in liquid nitrogen according to Table 1 (see step 3.1.1).Store the homogenized tissue at −70 °C.



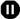

**PAUSE STEP** The homogenized samples can be stored at −70 °C for about 3 years.

3.Suspend 50 mg of homogenized tissue on ice in 0.5 mL of cold Lysis buffer 2 in 1.5 mL microcentrifuge tube.4.Centrifuge at 805× *g*, 4 °C for 5 min.5.Transfer the supernatant into new tubes.

#### 3.2.2. Protein A or Protein G Immobilization of Antibody

6.Use commercially available or made to order antibodies or serum that must meet the following requirements:●must recognize the native protein;●be suitable for immunoprecipitation.

The concentration of antibodies for immunoprecipitation is selected experimentally.

7.Choose magnetic beads with immobilized protein A or G particles based on the type of antibodies you have (according to the manufacturer’s recommendations).8.Wash the magnetic beads:●add 50 µL of suspension of selected magnetic beads to 200 µL of Binding buffer into a 2.0 mL microcentrifuge tube, mix gently;●using magnetic rack, let the particles settle on the magnet and carefully remove the supernatant with a pipette;●rewash the magnetic beads with 500 µL of Binding buffer using magnetic rack.9.Add 800 µL of Binding buffer, 8 µL of a Complete Protease Inhibitor Cocktail (×100) and preselected volume of antibody.10.Incubate the mixture at room temperature for 1 h with slow overhead rotation.



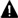

**CRITICAL STEP** During incubation prevent magnetic particles from sticking together or settle: in case of sticking, add more Binding buffer with the appropriate amount of protease inhibitors.

#### 3.2.3. Target Protein Immunoprecipitation

11.After the incubation time has passed, remove the supernatant and wash the beads 3 times with 500 µL of Binding buffer using magnetic rack.12.Transfer 150 µL suspension of magnetic beads into new 2.0 mL microcentrifuge tube.13.Add 350 µL of cell lysate with the prepared magnetic beads and incubate for 2 h at room temperature (or overnight at 4 °C) with slow overhead rotation on programmable rotator.



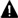

**CRITICAL STEP** During incubation prevent magnetic particles from sticking together or settle: in case of sticking, add more Binding buffer with the appropriate amount of protease inhibitors.

#### 3.2.4. Elution of Protein

14.Remove the supernatant and wash the beads 3 times with 500 µL of Binding buffer using magnetic rack.15.Transfer magnetic beads into new 1.5 mL microcentrifuge tubes and remove the supernatant.16.Add 100 µL of Elution buffer to the magnetic beads and incubate for 10 min at room temperature, stirring occasionally with finger taps.



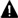

**CRITICAL STEP** Do not use boiling to elute fibrillar proteins. Boiling destroys amyloid fibrils.

17.When the elution time is over, select and transfer the supernatant into 0.5 mL microcentrifuge using magnetic rack.18.Add 13 µL of Neutralizing buffer to the first eluted fraction, mix gently by pipetting.19.Rinse magnetic particles with 50 µL of Elution buffer by gently pipetting.20.Select and transfer the supernatant to new 0.5 mL microcentrifuge tube.21.Add 6.5 µL of Neutralizing buffer to the second eluted fraction, mix gently by pipetting. Pool eluted fractions in one tube.

#### 3.2.5. Sedimentation of Fibrils by Ultracentrifugation

22.Transfer the eluted fraction into 230 µL ultracentrifuge tube.23.Centrifuge the probe at 436,000× *g*, 4 °C for 2 h.24.When centrifugation is over, remove the supernatant and dissolve the pellet (containing fibrils) in 10–20 µL of the deionized water.



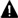

**CRITICAL STEP** The pellet may not be visible. Preliminarily mark with a marker the side of the ultracentrifuge tube where the sample will be deposited. If no pellet is visible, wash the bottom of the tube on the marked side with 10–15 µL of the deionized water.

25.Transfer the water solution containing fibrils into a 0.2 mL microcentrifuge tube and store at −20 °C.

#### 3.2.6. Analysis of Fibrils by Electron Microscopy

26.Adsorb 10 μL aliquot of fibril solution to the formvar coated copper grid for 1 min and remove.27.Wash twice with 10 μL of the deionized water for 1 min.28.Stain with 10 μL of 1% uranyl acetate for 2 min.29.Remove uranyl acetate and dry probe in air.30.Analyze fibrils with transmission electron microscopy.

#### 3.2.7. Analysis of Fibrils by Congo Red Staining and Polarization Microscopy

31.Spot 10 μL aliquot of fibril on a glass slide and dry.32.Apply 0.25% water solution of Congo red on top of the sample and incubate for 5 min at room temperature.33.After 5 min remove the staining solution and wash the sample with water to remove the rest of the unbound dye.34.Add 10 μL of water to the sample prior to covering with a cover glass.35.Analyze fibrils with polarized light microscope.

## 4. Expected Results

The combination of methods for the proteomic screening and isolation of amyloid fibrils makes it possible to find and identify amyloid proteins in different organisms. In particular, we identified by the proteomic approach the list of proteins that form amyloid-like aggregates in the brain of *Rattus norvegicus* [8]. Using fibril immunoprecipitation, we proved that one of these proteins, RNA-binding protein FXR1, functions in the brain of rats and other jawed vertebrates in the amyloid form [8,10]. In this paper, we present fibrils of the FXR1 protein isolated from the brain of the red-eared turtle as an illustration (Figure 3A). FXR1 fibrils isolated by immunoprecipitation are stained with Congo red and show yellow-green birefringence (Figure 3B,C).

We expect that similar results can be obtained using the described methodology for the detection of amyloids in any organisms with a sequenced and annotated genome. At least, using the PSIA-LC-MALDI method, amyloid-like proteins were identified in yeast, bacteria and pea seeds [6,7,9]. Moreover, the fibril immunoprecipitation method followed by Congo red staining provided direct evidence of the amyloid properties of yeast proteins Sup35 and Rnq1 in the strains containing [*PSI*^+^] and [*PIN*^+^] prions, respectively [19]. Slight differences in the protocols used for different organisms may be due to the peculiarities of tissue destruction and (or) the solubility of SDS-resistant protein aggregates. If necessary, for additional verification, fibrils isolated by immunoprecipitation can be stained with thioflavin S or T, as well as with conformation-dependent antibodies that recognize various amyloid fibrils and oligomers. Theoretically, the proteomic screening and immunoprecipitation of amyloid fibrils can be combined with quantitative mass spectrometry, but determination of the amount of protein in samples is not the task of this methodology.

In summary, the combination of proteomic screening for amyloids and the isolation of amyloid fibrils using immunoprecipitation opens broad perspectives for the search and identification of functional and pathological amyloids.

## 5. Reagents Setup

TBS. Add 800 mL of distilled water and a stir bar to a 1 L beaker, add 7.313 g NaCl. Stir until everything dissolves. Add 25 mL of Tris-HCl pH 7.6 and make volume up to 1 L with distilled water. TBS is stable at 4 °C for 3 months.Lysis buffer 1. To 9.58 mL of TBS buffer add 100 µL of 0.2 M PMSF, 200 µL of 0.5 M EDTA, 20 µL of 1 M DTT and 100 µL of a Complete Protease Inhibitor Cocktail. The buffer is ready to use. Store at 4 °C for 3 months.Lysis buffer 2. To 9.58 mL of PBS buffer add 100 µL of 0.2 M PMSF, 200 µL of 0.5 M EDTA, 20 µL of 1 M DTT and 100 µL of a Complete Protease Inhibitor Cocktail. The buffer is ready to use. Store at 4 °C for 3 months.PBS. Add 800 mL of distilled water and a stir bar to a 1 L beaker, add 8 g NaCl, 0.2 g KCl, 1.44 g Na_2_HPO_4_ and 0.245 g KH_2_PO_4_. Stir until everything dissolves. Adjust pH to 7.4 with 37% HCl and make volume up to 1 L with distilled water.25%-sucrose-TBS cushion. Dissolve 1.25 g sucrose in 4.5 mL TBS and mix. Dilute with TBS buffer to a volume of 5 mL. To prepare an SDS-containing sucrose cushion, the buffer solution should contain 500 µL of 10% SDS.Binding buffer. To 199.6 mL of PBS buffer add 40 µL of a Tween-20. The buffer is ready to use. Store at 4 °C for 3 months.Elution buffer. Add 50 mL of distilled water and a stir bar to a 100 mL beaker, add 9.4 g Glycine. Stir until everything dissolves. Adjust pH to 2.1 with 37% HCl and make volume up to 100 mL with distilled water.Neutralizing buffer. Add 400 mL of distilled water and a stir bar to a 500 mL beaker, add 181.71 g Tris. Adjust pH to 8.8 with 37% HCl and make volume up to 500 mL with distilled water.4× SDS-PAGE loading buffer. Mix 1 mL 100 mM Tris-HCl pH 6.8, 2 mL 2-Mercaptoethanol, 1 g SDS, 20 mg Bromophenol blue and 4 mL glycerol. Make volume up to 10 mL with distilled water.Tris buffer (pH 6.8). Add 400 mL of distilled water and a stir bar to a 500 mL beaker, add 181.71 g Tris. Adjust pH to 6.8 with 37% HCl or 0.1 M NaOH and make volume up to 500 mL with distilled water.Tris buffer (pH 7.6). Add 400 mL of distilled water and a stir bar to a 500 mL beaker, add 181.71 g Tris. Adjust pH to 7.8 with 37% HCl or 0.1 M NaOH and make volume up to 500 mL with distilled water.0.25% Congo Red. Dissolve 2.5 mg of Congo Red in 1 mL of distilled water. Filtrate using 0.22 µm Syringe Filter.1% uranyl acetate. Dissolve 5 mg of uranyl acetate in 0.5 mL of distilled water.

## Figures and Tables

**Figure 1 mps-06-00016-f001:**
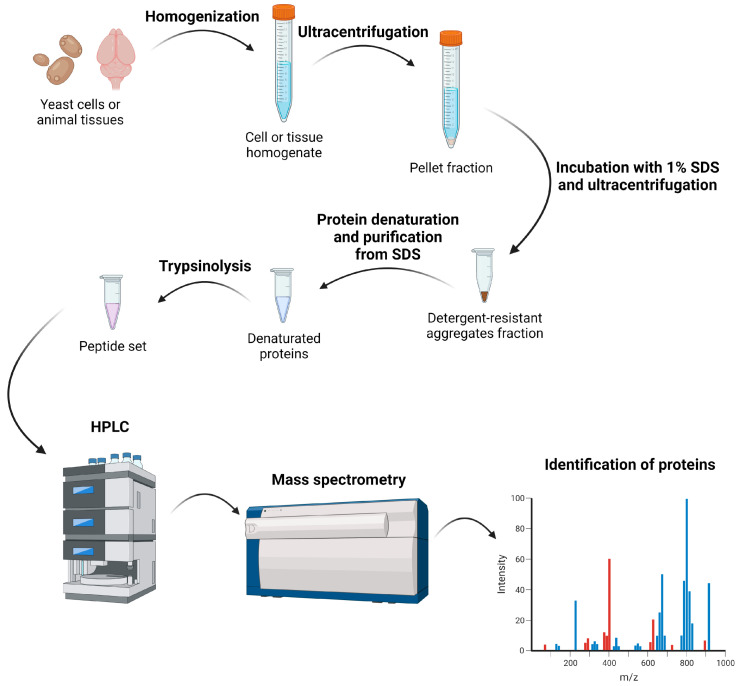
Scheme of proteomic screening for proteins forming SDS-resistant amyloid-like aggregates. Created with BioRender.com.

**Figure 2 mps-06-00016-f002:**
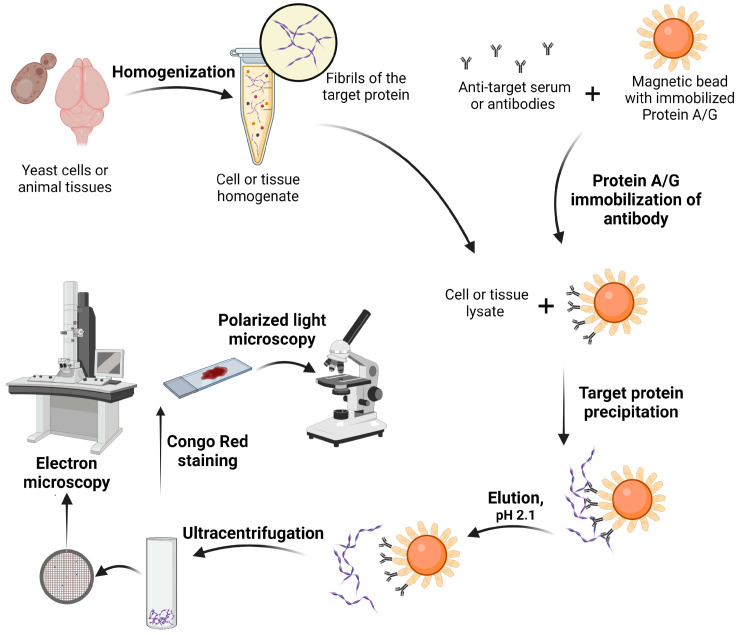
Scheme of the method of amyloid immunoprecipitation and ex vivo fibril staining. Created with BioRender.com.

**Figure 3 mps-06-00016-f003:**
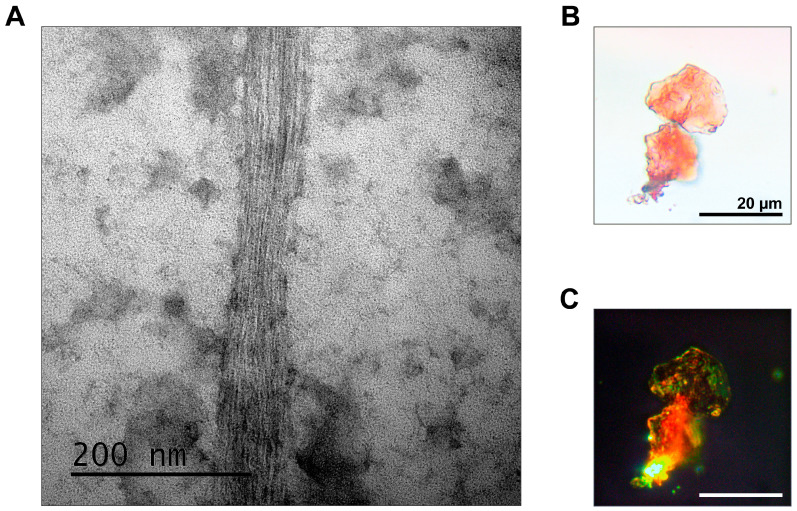
Analysis of the fibrillar structure and Congo red staining of the FXR1 protein isolated from the brain of *Trachemys scripta*. (**A**) FXR1 after immunoprecipitation detected by TEM; (**B**) Congo red staining analyzed in transmitted light; (**C**) Congo red staining analyzed in polarized light. Scale bars are presented.

**Table 1 mps-06-00016-t001:** Cycle conditions for sample homogenization.

Stage	Time	Cycle Number
Precooling	5 min	
Homogenization with a frequency of 15 cps	30 s	9 cycles
Cooling	2 min	

## Data Availability

Data sharing not applicable.

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
