# Peer review of "Search and Identification of Amyloid Proteins"

_mps, 2023, doi:10.3390/mps6010016_

Round 1

Reviewer 1 Report

This valuable procedure describes in detail a generally applicable method for identifying proteins in amyloid state in crude extracts of yeast or mammalian cells. As such, the protocol itself is a valuable contribution to the literature on natural amyloids, which are increasingly known to be important for regulation of cellular processes. 

The equipment, supplies, reagents, and procedures are well and clearly described.

Minor revisions:

Line 44: “… the universal feature of all known amyloid fibrils..” This statement requires a reference and/or qualification; “feature” or “general feature” is suggested.

Line 112: “…the prepared magnetic beads.” should be “…protein A- or G-coupled magnetic beads.”

 Line 116: “…and dissolved in water.” Should be “…resuspended in water.”

Line 166: Write out “Trifluoroacetic acid (TFA)…”

Line 330: “Dry the samples until completely removed formic…” Common English usage would be “Dry the samples until formic acid is completely removed…”

Line 340-342: Check that addition of DTT before iodoacetamide is a correct description. The amount described would inactivate 50% of the IA.

Author Response

Minor revisions:

Line 44: “… the universal feature of all known amyloid fibrils..” This statement requires a reference and/or qualification; “feature” or “general feature” is suggested.

The answer: We thank the Reviewer for the comment. Text is corrected (Lines 21 and 44). The references [4,5] were included in the text (Line 45).

Line 112: “…the prepared magnetic beads.” should be “…protein A- or G-coupled magnetic beads.”

The answer: Text is corrected (Lines 112 - 113).

 Line 116: “…and dissolved in water.” Should be “…resuspended in water.”

The answer: Text is corrected (Line 117).

Line 166: Write out “Trifluoroacetic acid (TFA)…”

The answer: Text is corrected (Line 166).

Line 330: “Dry the samples until completely removed formic…” Common English usage would be “Dry the samples until formic acid is completely removed…”

The answer: Text is corrected (Line 330).

Line 340-342: Check that addition of DTT before iodoacetamide is a correct description. The amount described would inactivate 50% of the IA.

The answer: Reducing sulfhydryl groups of cysteine with DTT allows better access of trypsin for the complete conversion of protein to peptide. IA is needed to modify the free cysteines after incubation with DTT. Thus, since DTT inactivates IA, we add it half as much as IA. This allows 50 mM IA to alkylate cysteine residues in the protein.

Reviewer 2 Report

The manuscript by Belashova and co-authors describes two methods, the combination of which makes it possible to find and identify amyloid proteins in the proteome of various organisms. The first method is Proteomic screening, the second is Immunoprecipitation staining.

The methods are clearly described. Moreover, the disadvantages of both approaches are well documented and stated. 

My only concern regards the final detections of the amyloid proteins in both methods. Due to my experience, I know that there are other possibilities to identify and also quantify amyloid proteins using targeted LC-MS approaches using instruments like QTRAP or QqQ in MRM or MRM3 mode. This technique can also provide quantitative data. Please have a look, at the following paper:

Kirmess, K.M., Meyer, M.R., Holubasch, M.S., Knapik, S.S., Hu, Y., Jackson, E.N., Harpstrite, S.E., Verghese, P.B., West, T., Fogelman, I. and Braunstein, J.B., 2021. The PrecivityAD™ test: Accurate and reliable LC-MS/MS assays for quantifying plasma amyloid beta 40 and 42 and apolipoprotein E prototype for the assessment of brain amyloidosis. Clinica Chimica Acta519, pp.267-275.

Pannee, J., Gobom, J., Shaw, L.M., Korecka, M., Chambers, E.E., Lame, M., Jenkins, R., Mylott, W., Carrillo, M.C., Zegers, I. and Zetterberg, H., 2016. Round robin test on quantification of amyloid-β 1–42 in cerebrospinal fluid by mass spectrometry. Alzheimer's & Dementia12(1), pp.55-59.

Iino, T., Watanabe, S., Yamashita, K., Tamada, E., Hasegawa, T., Irino, Y., Iwanaga, S., Harada, A., Noda, K., Suto, K. and Yoshida, T., 2021. Quantification of Amyloid-β in Plasma by Simple and Highly Sensitive Immunoaffinity Enrichment and LC-MS/MS Assay. The Journal of Applied Laboratory Medicine6(4), pp.834-845.

The sample preparation procedures of both methods are compatible with an MS-targeted approach. Thus, in my view, the authors should add a paragraph, probably after the "Expected results", calling it "Application Perspective". In this final paragraph, the authors could describe the possibility of coupling these sample preparation methods with a targeted LC-MS approach to reach reliable quantitative data, probably also adding a figure about the possible workflow. This can expand the applicability of the protocols and the interest of the scientists involved in amyloid protein identification and quantification. 

Author Response

The manuscript by Belashova and co-authors describes two methods, the combination of which makes it possible to find and identify amyloid proteins in the proteome of various organisms. The first method is Proteomic screening, the second is Immunoprecipitation staining.

The methods are clearly described. Moreover, the disadvantages of both approaches are well documented and stated. 

My only concern regards the final detections of the amyloid proteins in both methods. Due to my experience, I know that there are other possibilities to identify and also quantify amyloid proteins using targeted LC-MS approaches using instruments like QTRAP or QqQ in MRM or MRM3 mode. This technique can also provide quantitative data. Please have a look, at the following paper:

Kirmess, K.M., Meyer, M.R., Holubasch, M.S., Knapik, S.S., Hu, Y., Jackson, E.N., Harpstrite, S.E., Verghese, P.B., West, T., Fogelman, I. and Braunstein, J.B., 2021. The PrecivityAD™ test: Accurate and reliable LC-MS/MS assays for quantifying plasma amyloid beta 40 and 42 and apolipoprotein E prototype for the assessment of brain amyloidosis. Clinica Chimica Acta519, pp.267-275.

Pannee, J., Gobom, J., Shaw, L.M., Korecka, M., Chambers, E.E., Lame, M., Jenkins, R., Mylott, W., Carrillo, M.C., Zegers, I. and Zetterberg, H., 2016. Round robin test on quantification of amyloid-β 1–42 in cerebrospinal fluid by mass spectrometry. Alzheimer's & Dementia12(1), pp.55-59.

Iino, T., Watanabe, S., Yamashita, K., Tamada, E., Hasegawa, T., Irino, Y., Iwanaga, S., Harada, A., Noda, K., Suto, K. and Yoshida, T., 2021. Quantification of Amyloid-β in Plasma by Simple and Highly Sensitive Immunoaffinity Enrichment and LC-MS/MS Assay. The Journal of Applied Laboratory Medicine6(4), pp.834-845.

The sample preparation procedures of both methods are compatible with an MS-targeted approach. Thus, in my view, the authors should add a paragraph, probably after the "Expected results", calling it "Application Perspective". In this final paragraph, the authors could describe the possibility of coupling these sample preparation methods with a targeted LC-MS approach to reach reliable quantitative data, probably also adding a figure about the possible workflow. This can expand the applicability of the protocols and the interest of the scientists involved in amyloid protein identification and quantification. 

The answer: We agree with the reviewer that quantitative mass spectrometry could potentially be used in proteomic screening of amyloids. This information is added in manuscript (L. 491 – 494). However, the reviewer writes about the quantitative analysis of already known amyloids using targeted mass spectrometry methods. Our proteomic screening is used to detect and identify amyloids that have not been previously characterized. In this case, targeted mass spectrometry is not applicable, since we are looking for unknown proteins that have amyloid properties. It is possible to estimate the amount of such proteins in various samples, but this is not the purpose of the methods developed by us.